

# A new approach to understanding fluid mixing in process-study models of stratified fluids

Samuel George Hartharn-Evans[1], Marek Stastna[2], and Magda Carr[1]

[1]School of Mathematics, Statistics and Physics, Newcastle University, United Kingdom
[2]Department of Applied Mathematics, University of Waterloo, Waterloo, Canada.

**Correspondence:** Samuel G. Hartharn-Evans (s.hartharn-evans2@ncl.ac.uk)

**Abstract.** While well established energy-based methods of quantifying diapycnal mixing in process-study numerical models are often used to provide information about when mixing occurs, and how much much mixing has occurred, describing how and where this mixing has taken place remains a challenge. Moreover, methods based on sorting the density field struggle with under resolution and uncertainty as to the definition of the reference density when bathymetry is present. Here, an alternative method of understanding mixing is proposed. Paired histograms of user selected variables (which we abbreviate USP) are employed to identify mixing fluid, and are then used to identify regions of fluid in physical space that are undergoing mixing. This paper presents two case studies showcasing this method: shoaling internal solitary waves and a shear instability in cold water influenced by the nolinearity of the equation of state. The USP method identifies differences in the mixing processes associated with different internal solitary wave breaking types, including differences in the horizontal extent and advection of mixed fluid. The method is also used to identify how density, and passive tracers are mixed within the core of the cold-water Kelvin-Helmholtz instability.

## 1 Introduction

At the largest scales, the ocean is stably stratified, typically with warmer, less dense water overlaying cold, and more dense water (although in many locations, such as the polar regions the stratification is salinity dominated instead and warm water can underlay cold). Similar density stratifications also exist across other geophysical settings, such as lakes, estuaries, and the atmosphere, and the process by which the layers mix is of great interest to a range of disciplines, from biological interests in the distribution of nutrients, plankton, and sediment, to physical oceanography interests in the global (and local) distribution of heat and buoyancy. This mixing primarily occurs by flows in these fluids leading to small scale motions that stir the fluid and stretch density interfaces. Across the increased surface area, molecular diffusion is effectively sped up in the process of diapycnal mixing. However, mixing processes occur at scales much smaller than oceanic and atmospheric models grid scales. Process-scale numerical modelling is a useful tool to better understand the routes to mixing, with a view to improving the way that climate-scale and regional models parameterise such processes through turbulent or eddy diffusivity. Large scale models may use sophisticated parameterisations of eddy diffusivity, dependent on local velocity shear and buoyancy, and examples include the well-known $k - \epsilon$ model for RANS (Pope, 2000) and the Samgorinsky model for LES (Wyngaard, 2010).



The mechanical process of stirring is a geometric deformation of fluids, and does not in itself imply irreversible mixing. However, the presence of a diffusion term in the equations of motion (equation 5) allows mixing (where the concentration of a tracer, or density, of a given fluid parcel is modified) to occur across density gradients that are stretched by stirring. Commonly, mixing in numerical simulations is quantified according to the framework set out by Winters et al. (1995). Under this framework, energy is partitioned into various components; Kinetic Energy (KE), Background Potential Energy (BPE), Available

Potential Energy (APE) and Internal Energy (IE). APE is the potential energy available to be released to kinetic energy. APE is computed by first adiabatically redistributing the density field to find the lowest possible energy state (represented by the BPE). The conversion between APE and BPE is considered the energy used to conduct irreversible diapycnal mixing, and is therefore an important concept. Crucially, calculating APE involves sorting the density field in order to identify the lowest energy state achievable by adiabatic redistribution (or under certain formulations, a far-field reference density profile (Lamb, 2008)). Other

formulations for measuring mixing have been suggested, such as computing the Thorpe length scales (Thorpe, 1977), which have been applied to both laboratory experiments and numerical simulations (e.g. Carr et al., 2017).

    The most widely used APE framework is valuable due to its uses in comparisons to the field, and parameterisation in larger scale models. However, the Winters et al. (1995) sorting process does little to inform us of interesting questions around where and how this mixing is taking place, or what happens to the mixed fluid (Moum et al., 2003; Carr et al., 2017). Exploration of

the spatial distribution BPE density, and APE density, or local APE can give us some further information about the flow (e.g. Lamb, 2008; Scotti and White, 2014), but are rarely considered in comparison with the domain integrated, or bulk, values. This is in part due to the complexity of calculating these local values.

    The diapycnal mixing of passive tracers was considered from a different point of view by Penney et al. (2020) using simple two-variable probability density histograms showing the statistical relationship between a tracer and density. Penney *et al*

referred to their primary tool as "weighted density-tracer scatter plots", since we propose a methodology that is more general, we will adopt the acronym USP, for user-controlled scatter plot. The user specifies the two variables chosen, and the manner in which their ranges are set in order to focus on dynamical phenomena of interest. The pair of fields studied in detail in Penney et al. (2020) is a subset of our more general methodology.

    To demonstrate the efficacy of our methodology we choose two application areas, one in which the geometry is complex

(shoaling internal waves) and one in which the equation of state is nonlinear (in water below the 4 °C temperature of maximum density). Grace et al. (2021) simulated the evolution of gravity currents in water below the 4 °C temperature of maximum density. They demonstrated profound asymmetries between cold gravity currents intruding into warm water, and warm gravity currents intruding into cold water. They labelled this temperature regime the "weak cabbeling" regime, because two water parcels mixed together yield a different density than the average density of the individual parcels. Grace *et al* found histograms

of a single fluid quantity (e.g. temperature) to be a useful analysis tool in characterizing the gravity currents.

    Here, we bring the methods of Penney et al. (2020) and Grace et al. (2021) together, investigating the utility of selecting fluid parcels based on the combination of two quantities, for example asking where the density is within one range, and the kinetic energy (KE) exceeds a threshold value simultaneously. An interactive Matlab tool is developed, and applied to two example flows which have previously been studied extensively. Firstly shoaling nonlinear Internal Solitary Waves (ISWs), that result



from such waves propagating up-slope are studied, a process investigated in numerous process-studies for its role in transport and mixing of heat, nutrients, and sediment (e.g. Michallet and Ivey, 1999; Aghsaee et al., 2010; Sutherland et al., 2013; Arthur et al., 2017; Hartharn-Evans et al., 2022a). Secondly, the stratified shear flow and its Kelvin Helmholtz instability, a feature explored in many studies relating to mixing due to its fundamental importance (e.g. Winters et al., 1995; Caulfield and Peltier, 2000; Peltier and Caulfield, 2003; Caulfield, 2021), and here applied in three dimensions to a cold water setting, where the

nonlinear equation of state alters the dynamics. Such features have been observed in a range of environmental settings, from enhancing mixing by ISWs on the Oregon Shelf (Moum et al., 2003), deepening the Arctic Ocean surface mixed layer (Lincoln et al., 2016), and affecting the highly stratified Connecticut River estuary (Geyer et al., 2010).

The joint probability histograms, and specifically their change in their form over time, allows identification of fluid quantities that are interesting. Additionally, the method allows us to determine the physical region of fluid where these quantitatively

unambiguously identified "interesting things" are happening. Instead of the understanding of how much bulk mixing is taking place (as provided by the sorting algorithm), we can understand where, when and perhaps how mixing is taking place. Using this method of selecting regions based on user-selected paired histograms (USP) also has advantages over the Winters et al. (1995) sorting algorithm in that it is unaffected by small perturbations inherent to numerical methods (e.g. due to uncertainties in bathymetry, or general under resolution), and as there is no assumption of mass (or energy) conservation inside the region

to be analysed, the spatial domain for analysis can be restricted to just the physical area we're interested in.

This paper is organised as follows. In §2.1, the USP methodology is introduced. In §2.2 the numerical model SPINS is introduced, with specific model setups for the fissioning ISW and cold shear flows are introduced in §§2.2.1, 2.2.2 respectively. The method is then applied to identify the causes of different mixing regimes according to wave breaking types in §3.1, followed by application of the method to understand diapycnal mixing in the cold shear instability in §3.2. The results conclude with the

introduction of a passive, rather than active tracer to the cold shear instability §3.3, identifying the differences between active and passive tracers with differing diffusivity.

## 2 Methods

### 2.1 Paired Histograms

This study employs pseudocolour plots showing the characteristics of fluid parcels in a two-dimensional (2-D) space where

the two dimensions, instead of being spatial dimensions, are chosen fluid characteristics, similar to the weighted density-tracer scatter plots in Penney et al. (2020). Each USP plot shows the characteristics of fluid parcels in terms of this redefined 2-D space, with two fluid properties as their coordinates, and the colour showing the proportion of the fluid within each discretized coordinate bin. The change of a scatter plot, reminiscent of temperature-salinity plots widely used to trace water masses in oceanography (Helland-Hansen, 1916; Imasato et al., 1993), to the paired histogram by discretising into bins and applying

weightings is shown in figure 1, and described here.





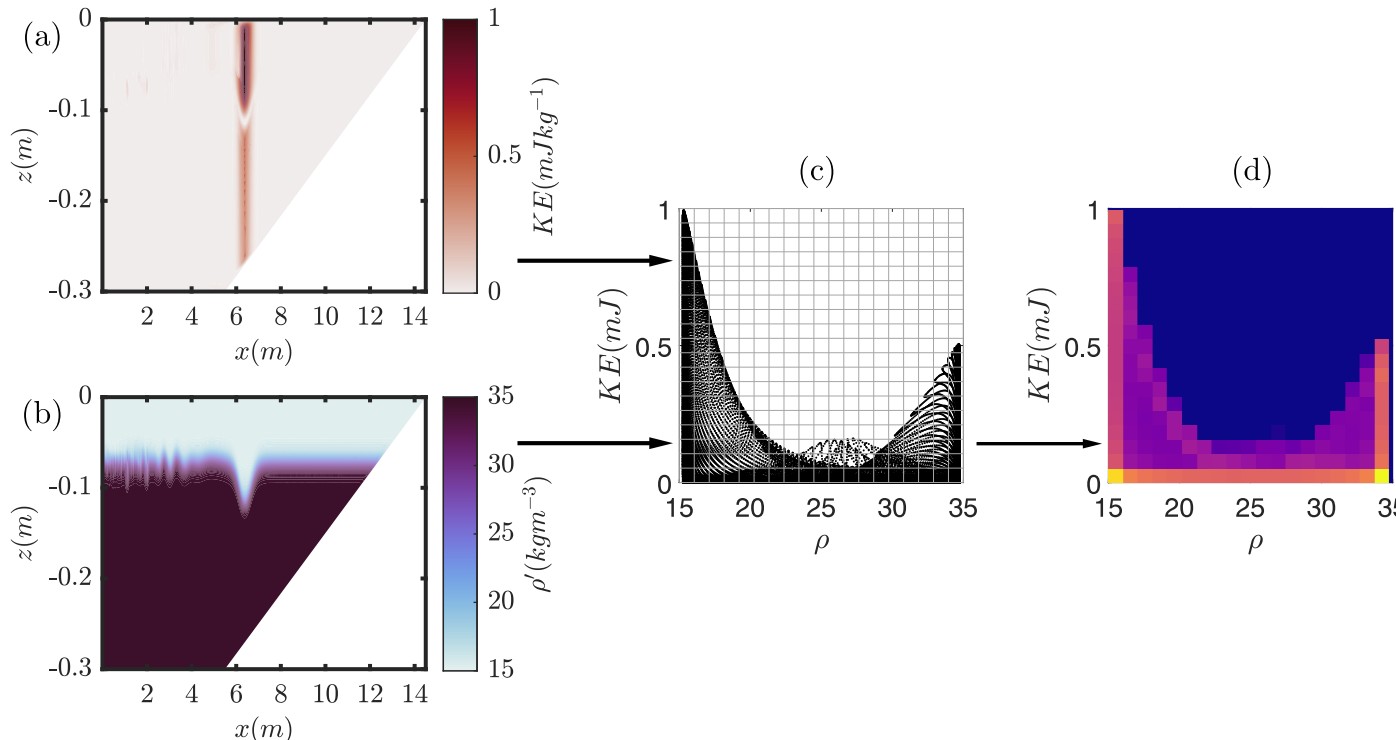

**Figure 1.** Schematic of the creation of USP diagrams. The combination of two fluid properties (left) are represented in variable-variable space (centre), and then summed into discrete bins (right)

Following the method of Penney et al. (2020), the algorithm employed is applied as follows. For two variables $\phi$ and $\theta$ (which could for example represent density and kinetic energy), the domains are subdivided into $N_\phi$ and $N_\theta$ bins, with sizes:

$$\delta\phi = \frac{\phi_{max} - \phi_{min}}{N_\phi}, \qquad\qquad \delta\theta = \frac{\theta_{max} - \theta_{min}}{N_\theta} \qquad\qquad (1)$$

For each grid cell, the nearest bin centre is identified:

$$\phi_i = \phi_{min} + \frac{2i-1}{2}\delta\phi, \quad i = 1, 2, ..., N_\phi, \qquad\qquad \theta_j = \theta_{min} + \frac{2j-1}{2}\delta\theta, \quad j = 1, 2, ..., N_\theta \qquad\qquad (2)$$

For each grid cell, the $I_{ij}(\phi, \theta)$ is calculated:

$$I_{ij}(\phi, \theta) = \begin{cases} dxdzdy, & (\phi(x,z) - \phi_i, \theta(x,z) - \theta_i) \in \left[-\frac{1}{2}\delta\rho, \frac{1}{2}\delta\rho\right] \times \left[-\frac{1}{2}\delta\theta, \frac{1}{2}\delta\theta\right] \\ 0, & \text{otherwise}, \end{cases} \qquad (3)$$

and the total weight for each given bin with centre $(\phi_i, \theta_j)$ is given by:

$$W_{ij} = \frac{1}{V}\sum I_{ij}(\phi, \theta). \qquad\qquad (4)$$





This is that for a given bin the weight $W_{ij}$ is the total volume of grid cells that had $(\phi_i - \frac{1}{2}\delta\phi \leq \phi < \phi_i + \frac{1}{2}\delta\phi)$ and $(\theta_j - \frac{1}{2}\delta\theta \leq \theta < \theta_j + \frac{1}{2}\delta\theta)$. The end result being a bivariate weighted histogram, where the weighting ensures the Probability Density Function (PDF), relates to the probability a volume of fluid has given properties in mapped cases. A code for this applied to the numerical model SPINS (introduced in §2.2) can be found at: https://github.com/HartharnSam/SPINS_USP, which generalises the formulation in Penney et al. (2020) to any two fluid variables in a non-uniform grid.

The new tool introduced here are Region of Interest (ROI) plots. Here, the variable is plotted in physical space, but only where conditions identified on the paired histogram are met, for example that $1040 < \rho < 1050 \text{ kgm}^{-3}$ and $0.1 < KE < 0.2 \text{ Jkg}^{-1}$. In relation to mixing in the shoaling wave simulations, the kinematic quantity investigated is enstrophy density, $\Omega = \frac{1}{2}|\boldsymbol{\omega}|^2$ (where $\omega = \nabla \times \boldsymbol{u}$) which describes the rotational energy linked to dissipation in 2-D flows, and so is a quantity associated with (but not a measure of) mixing. The Kinetic Energy density, $KE = \frac{1}{2}\boldsymbol{u}^2$ is used as a measure of energy of translation in the flow.

In order to unambiguously identify time periods of interest in the various case studies reported on below we have utilised the methodology of Shaw and Stastna (2019). Briefly, this technique computes the Empirical Orthogonal Functions (EOFs) for a particular physical quantity using the built in *svds* function in Matlab. The EOFs are used to create reconstructions of the original physical field, and the infinity norm of the difference between the reconstruction and original field is plotted as a function of time and the number of modes used in the reconstruction. Time periods for which a jump in the number of EOFs needed to obtain an approximation below a given error tolerance are taken as "periods of interest" and hence selected for detailed examination. This technique is attractive because it builds on a standard technique (EOFs), while addressing its well-known shortcoming (the fact that individual EOFs are not physical quantities).

## 2.2 Numerical model

Simulations were carried out with the pseudospectral code SPINS described in Subich et al. (2013). The code has been thoroughly validated using theoretical result and physical laboratory experiments in a number of different configurations including, shear instabilities, boundary layer instabilities (e.g. Harnanan et al., 2017), interaction with topography (e.g. Deepwell et al., 2017), and shoaling ISWs (e.g. Hartharn-Evans et al., 2022a). It is available for download through its online manual:

```
https://wiki.math.uwaterloo.ca/fluidswiki/index.php?title=SPINS_User_Guide
```

The model solves the stratified Navier–Stokes equations subject to the Boussinesq approximation:

$$\frac{\partial \boldsymbol{u}}{\partial t} + \boldsymbol{u} \cdot \boldsymbol{\nabla} \boldsymbol{u} = -\frac{1}{\rho_0} \boldsymbol{\nabla} P + \nu \nabla^2 \boldsymbol{u} - \frac{\rho g}{\rho_0} \hat{k}, \tag{5}$$

$$\boldsymbol{\nabla} \cdot \boldsymbol{u} = 0, \tag{6}$$

$$\frac{\partial \rho}{\partial t} + \boldsymbol{u} \cdot \boldsymbol{\nabla} \boldsymbol{\rho} = \kappa_\rho \nabla^2 \rho, \tag{7}$$

$$\frac{\partial \Xi_i}{Dt} + \boldsymbol{u} \cdot \boldsymbol{\nabla} \Xi_{\boldsymbol{i}} = \kappa_{\Xi_i} \nabla^2 \Xi_i \tag{8}$$




where $\boldsymbol{u}$ is the velocity, $t$ is time, $P$ is the pressure, $\Xi_i$ is the passive tracer field, $\rho$ is the density and $\rho_0$ is some reference density of the fluid, and throughout this paper density is reported using the density anomaly $\rho' = \rho - \rho_0$. The physical parameters are gravity $g$ (set at $9.81 \mathrm{ms}^{-2}$), the shear viscosity $\nu$ (set at $10^{-6} \mathrm{m}^2 \mathrm{s}^{-1}$, chosen to be consistent with the physical value) and scalar diffusivity $\kappa_\rho$ (or $\kappa_{\Xi_i}$ for tracer). The unit vector in the vertical direction is denoted by $\hat{k}$. The boundary conditions, domain dimensions and other varying parameters for each simulation are described for each case in §2.2.1 and 2.2.2.

### 2.2.1   Shoaling Internal Solitary Waves

To illustrate the USP method, simulations from previously published papers on ISW shoaling will be utilised, namely simulations 27_111120 (here the surging case), 26_091120 (here the collapsing case), 24_071020 (here the plunging case) from Hartharn-Evans et al. (2022a) and the Thin-20L case (here the fissioning case) from Hartharn-Evans et al. (2022b). Full details of the simulations can be found in the respective papers. ISWs were simulated in a rectangular tank with waves initiated using

the lock gate technique, and a slope at one end of the tank as shown in the schematic of the model set-up in figure 2 (a). The length of the tank $L_x = 7$ m (extended to $L_x = 14.5$ m for the fissioning case), and the depth of the tank $L_z = 0.3$ m. A hyperbolic smoothing function produces the gate region by a numerical step in density $0.3$ m from the left end of the tank, from which an ISW is produced and propagates from left to right.

  No slip boundary conditions were applied at the flat upper, and mapped lower boundaries to satisfy model requirements. A

mapped Chebyshev grid is employed in the vertical, implying a clustering of points near both the upper and lower boundary that scales with the number of points in the vertical squared, and that vertical resolution improves over the slope. Free-slip boundary conditions were applied at the vertically oriented left and right ends of the computational domain, the grid spacing of which was regularly spaced. Grid resolution was 4096 points in the $x$ and 256 grid points in the $z$ coordinate, giving $dx = 1.7$ mm in all cases except the fissioning case, where $dx = 3.5$ mm. Away from the slope, $dz$ varies between $0.124$ mm at the upper and

lower boundaries and $1.8$ mm near mid-depth, simulations were two-dimensional and $\kappa_\rho = 10^{-7} \mathrm{\ m}^2 \mathrm{s}^{-1}$.

  The vertical density profile in the main tank was set according to the smoothed two-layer, or hyperbolic tangent profile:

$$\rho(z) = \rho_0 + \frac{\Delta\rho}{2} \tanh\left(\frac{z - z_{pyc}}{h_{pyc}}\right), \tag{9}$$

set for a system like that studied previously in the literature consisting of a thin, linearly stratified pycnocline ($h_{pyc} = 0.015$ m) sandwiched between homogeneous layers, here referred to as "thin tanh stratification". The density change, $\Delta\rho = 20 \mathrm{\ kgm}^{-3}$,

the reference density $\rho_0 = 1026 \mathrm{\ kgm}^{-3}$ and the depth of the pycnocline, $z_{pyc} = 0.07$ m. Simulations are carried out for a slope of $s = 0.2$ in all cases except the fission case at $s = 0.033$, and in all cases the slope height is $0.3$ m, the full depth of the water column. The bottom boundary follows the form of Lamb and Nguyen (2009):

$$z = s \left( \mathrm{itanh}(x, L_x - L_s, d) - \mathrm{itanh}(x, L_x, d) \right) \tag{10}$$

where

$$\mathrm{itanh}(x, a, d) = \frac{1}{2}\left( x - a + d \ln\left( 2 \cosh\left(\frac{x - a}{d}\right) \right) \right) \tag{11}$$



and $d \, (= 0.03)$ represents a characteristic distance for the transition from $0$ to a constant slope of $1$, and $L_s$ the length of the slope ($1.5$ or $9$m for the fission case). The function smooths the transition from the flat bed to the slope, which is necessary for the spectral code. The surging ($a = 0.009$ m), collapsing ($a = 0.048$m) and plunging ($a = 0.076$ m) cases represent increasing wave amplitudes ($a$), and the fissioning ($a = 0.063$m) case has an intermediate wave amplitude.

### 2.2.2 Cold shear instability

A Kelvin-Helmholtz instability was produced for a three-dimensional (3-D) simulation in a domain $0.512$ m in the $x$ direction, and $0.128$ m in the $y$ and $z$ directions by initialising a temperature stratified shear flow perturbed with white noise. Free slip boundary conditions were applied at the upper and lower boundaries, with regular grid spacing in all dimensions, and periodic boundary conditions in the $x$ and $y$ domains. To produce a stratified shear flow, the temperature field, $T(z)$ and horizontal velocity field, $u(z)$ were initialised as follows:

$$T(z) = T_0 + \frac{\Delta T}{2}\left(1 - \tanh\left(\frac{z - z_{mix}}{h_{mix}}\right)\right), \qquad u(z) = \Delta u \tanh\left(\frac{z - z_{mix}}{h_{mix}}\right). \qquad (12)$$

A nonlinear equation of state suitable for cold, fresh water was chosen. The equation of state to calculate densities from temperature and salinity is the polynomial fit from Brydon et al. (1999), assuming salinity and excess pressure are $0$ everywhere. The background temperature and velocity gradient regions are thus in a similar location, but the density stratification is asymmetric (figure 2 b). Salinity was set to $0$ everywhere, and $v, w$ were set at rest, with small white noise perturbations to initialise the instability. The interface depth, $z_{mix}$ and thickness, $h_{mix}$, were $0.064$m and $0.01$m respectively. The minimum initial Richardson number, $Ri_0$ is defined as:

$$Ri_0 \approx \frac{g \Delta \rho h}{\rho_0 (\Delta u)^2}. \qquad (13)$$

The configuration was set such that $Ri_0 \ll 0.25 \; (= 0.087)$. The low value was chosen because $Ri = 0.25$ is the necessary, but not sufficient, criterion for shear instability. For this temperature range, the density range is very small, so for the cold shear instability cases, the density anomaly, $\rho'$ is a useful measure.

Such flows characterise situations where the temperature and momentum are linked, rather than density and momentum. For example, a warm river flow entering a cold lake.

To establish the interplay between passive and active tracers, a passive tracer (i.e. dye in a laboratory setting) was additionally initialised in two further simulations with the same initial conditions as temperature (equation 12), but with each case performed at different tracer diffusivity to the temperature tracer, as described in §3.3.

## 3 Results

### 3.1 Fissioning Internal Solitary Waves & the newly-mixed layer

The evolution of shoaling ISWs in this stratification takes one of four forms: surging, collapsing, plunging or fissioning (for more details see e.g. Sutherland et al., 2013; Hartharn-Evans et al., 2022a). Each of these manifests different mixing and





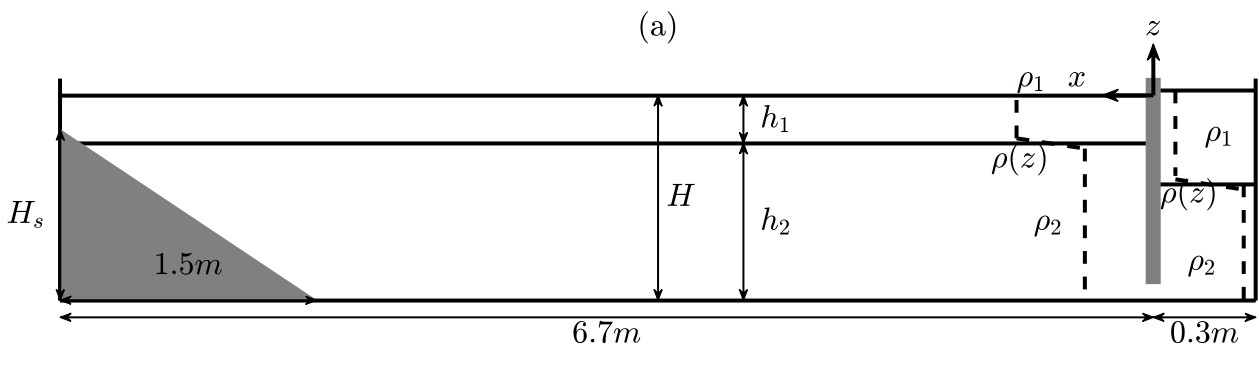

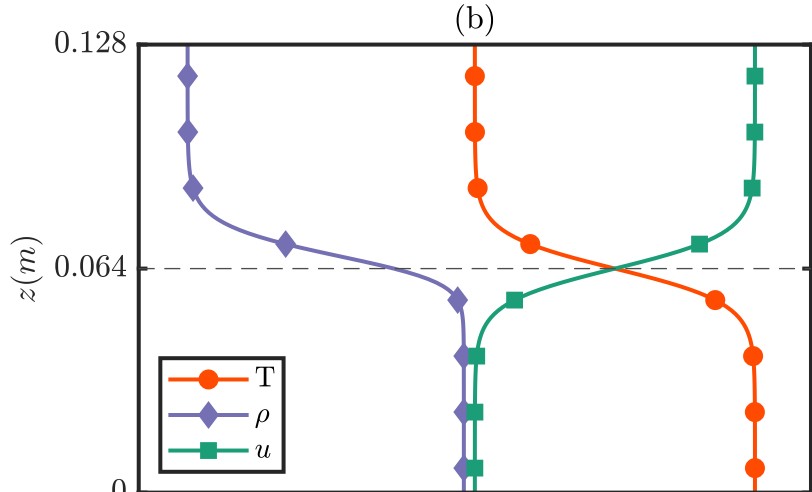

**Figure 2.** (a) Schematic diagram of numerical domain used for the shoaling internal solitary waves simulations in this study. (b) Schematic initial vertical profiles of the density, temperature, and velocity fields for the cold shear instability simulations (horizontal scales adjusted), the dashed horizontal line indicates the mid-tank, and centre of the interface for $T$ and $u$.

dissipation characteristics, as identified via bulk mixing measures (Arthur and Fringer, 2014). Figure 3 shows the evolution of one of these examples, collapsing, which occurs with moderate steepness waves over moderate slopes, will be explored here in more detail (each of the other examples is presented in Supplementary Figures 1-3). A more complete description of these processes can be found in Hartharn-Evans et al. (2022a) §4.3, but here the focus is on the evolution of USP during this process.

First, the reference USP case for an ISW prior to interacting with the slope, as shown in figure 1, is explored. At this point, most of the fluid in a three-layer stratification is at either extreme of density ($\rho = 1015$ or $1035\mathrm{kgm}^{-3}$) (figure 1 b, d), with only a small amount of fluid in intermediate densities across a thin pycnocline. Meanwhile, the KE is strong in both the upper and lower layers around the wave itself (figure 1 a), and is small at the centre of the pycnocline due to shear here. The overall result is an arcing pattern in USP space. This pattern is asymmetric, due to the asymmetric stratification, which concentrates the KE in the upper layer. In order to investigate mixing due to the shoaling of this wave, we will now explore how this pattern







**Figure 3.** Time sequence of collapsing wave shoaling process shown as $\rho'$ (top), $\Omega$ (middle), and USP for the same variables (bottom). $t = [55, 60, 70, 75]$s



changes during the shoaling process, and identify the fluid regions which these new fluid properties apply to, instead using enstrophy, $\Omega$ as a measure of energy in the flow available for mixing. Almost inverse to the KE USP in figure 1, $\Omega$ is highest at the highest shear region across the pycnocline in the wave, and at the upper and lower boundaries around the wave (figure 3 i).

As the wave begins to interact with the slope and steepen (figure 3 a, b, e, f), a separation bubble forms on the slope due
to boundary layer separation Carr et al. (2008); Boegman and Ivey (2009), meanwhile enstrophy remains constrained to the pycnocline (now extended), and the separation bubble, reflected in much higher enstrophy across mid-densities (figure 3 f, j). As the wave continues to propagate upslope and concentrate into a shallower region, the separation bubble strengthens and produces a global instability (figure 3 c, g). USP reveals higher enstrophy primarily associated with higher densities, with remenants of the high enstrophy at the upper and lower boundaries still visible in USP as the high enstrophy at high and low
density limits (figure 3 g, k). This evolves into a bolus of fluid moving upslope (figure 3 d, h). Although plots of enstrophy and density indicate mixing is occurring around the pycnocline at the head of the bolus (figure 3 d, h), USP is informative of where mixing is occurring. As the bolus propagates upslope, the skew of high enstrophy fluid towards higher densities (lower layer fluid) becomes clearer in the USP diagram (figure 3 l).

A region of USP space can be identified at early output times, which includes only $\Omega$ higher than that associated with a
stable ISW, and intermediate densities, and therefore is here taken to represent active mixing regions (white box figure 3 i). The fluid that has the properties contained within such a region can be identified, and is plotted in figure 4 e-h, for each type of wave shoaling. For a collapsing wave at a late time step, this shows a large continuous patch of actively mixing fluid, extending over around $0.7\mathrm{m}$ (figure 4 f), concentrated at intermediate densities (the pycnocline), although slightly skewed towards higher densities (figure 4 j).

Following this same process for four example waves across the four breaking regimes reveals their mixing properties. Surging is observed for small amplitude waves, and the resulting process is a non-turbulent surge of dense fluid propagating upslope (figure 4a). A single region of high enstrophy forms as the combination of that at the pycnocline and the lower boundary (figure 4 e), and this is advected upslope. Little mixing occurs during this process (figure 4 i), and the small active mixing region is advected upslope in the pulse of dense fluid. Plunging, which takes place at high slope and wave steepness values, results in
an anvil-shaped instability plunging forward from the rear steepening face of the wave. As a result, overturning occurs, and this process has been associated with high levels of mixing. Although this process produces high enstrophy across the density range, indicating a lot of mixing (figure 4 k), the actively mixing fluid is less continuous, and stretches over a smaller horizontal extent than a collapsing wave (figure 4 c, g). The active mixing region is vertically extensive, reflected in the USP plot with high enstrophy across all densities, but a slight skew in the USP diagram indicates mixing is associated with lower densities mixing
to intermediate densities rather than higher densities to intermediate densities. The final wave shoaling behaviour, fissioning, is shown in the right column of figure 4, and occurs over gentle slopes. The formation of multiple dense pulses of fluid (figure 4 d), which in this example are quasi-turbulent, produces actively mixing fluid over a long horizontal extent (figure 4 h), with active mixing occurring as the pulses propagate upslope. A strong skew in the USP indicates mixing is primarily associated with high densities, and so the lower layer (figure 4 l). This follows with the concept of dense pulses propagating upslope and
mixing as they do so.





**Figure 4.** Late time step density (top) region of interest (ROI, middle) and USP (lower) plot for four example waves, representing different breaking regimes. The white box in the lower panels indicate the ROI selected, based on USP at earlier time steps.



As an overview of this pattern, as the wave steepness increases, and therefore the wave breaking regime moves from surging through collapsing to plunging, the region of actively mixing fluid increases. Fissioning takes this a step further, with a similar input wave to the collapsing case, the gentle slope allows the wave to mix over a longer distance.

### 3.2 Cold shear instability

The shear flow undergoes the typical evolution through the formation of Kelvin-Helmholtz (K-H) instability, as follows. Initially, the flow is a shear flow in stably stratified fluid (figure 5 a, e), where the Kinetic Energy (KE) is high in both the upper and lower layers, but at the interface of the layers is zero, at the inflection point of the horizontal velocity (figure 2 b). As the fluid is a two layered shear flow, one would expect the $\rho'/KE$ USP to be a symmetrical curve, where the highest kinetic energy is associated with the highest and lowest kinetic energies away from the interface (which is also where most of the fluid

resides), whilst intermediate densities represent the shear layer where KE is lower. Here, the $\rho'/KE$ USP show a "crooked smiley face", with most the fluid concentrated at the extremes of density, but due to the temperature stratification and non-linear Equation of State, the curve is shifted towards higher densities (figure 5 i). Despite kinetic energy being high at this stage in the other directions, there is very little kinetic energy in the $v$ direction ($\mathrm{KE}_v = \frac{v^2}{2}$), indicating the flow features at this stage are not dependent upon 3-D elements of the flow (figure 5 i, m).

It is worth noting that while the evolution of the shear instability is generic, the nonlinearity of the equation of state means that the symmetry across the center of the co-located shear and density layers is broken. The visual manifestation of the billows is effectively slightly biased toward one side.

Small fluctuations in the flow result in stationary waves growing on the interface, which due to the shear, grow, and roll up forming braids and billows (figure 5 b, f). As the billows form and stretch, an increasing amount of the flow's KE builds around

the billows, and therefore associated with the intermediate densities (figure 5 f, j). Until now, the flow is two-dimensional, with negligible kinetic energy in the transverse direction (figure 5 m, n), and little transverse variability. During the braiding and billow growth, the pycnocline is stretched thin (figure 5 b), enhancing the potential for mixing by stretching density gradients. The roll up of these billows is such that alternating layers of high and low density form, with the density now statically unstable, leading to turbulence (and in turn spanwise instability) and mixing. These billows pair and coalesce by $t = 225$ s (figure 5 c).

The kinetic energy associated with intermediate densities continues to build as the billows coalesce, but by this stage, an increased volume of fluid that represents fluid in the $\rho' = -0.03$ to $-0.01\mathrm{kgm}^{-3}$ range is evident (figure 5k), indicating that mixing has occurred in the fluid. When the billows coalesce, $\mathrm{KE}_v$ is high, and almost all of this is associated with the same density range as that of the newly mixed fluid layer at late stages (figure 5 o). This indicates the importance of these transverse flow features in the mixing produced by billows formed from a K-H instability. The skew in the density of newly-mixed fluid,

as a result of the cold temperature nonlinearity in density, is to some extent visible (but easily missed) in the $\rho'$ plots (figure 5 c), but is only truly apparent in USP space (figure 5 k, l, o, p).

Beyond this time step, mixing across the braids continues, and leads to a new state (figure 5 d) where a quasi-turbulent intermediate density layer forms, which represents a new steady state, where the $Ri$ is reduced by a smoothed density gradient.



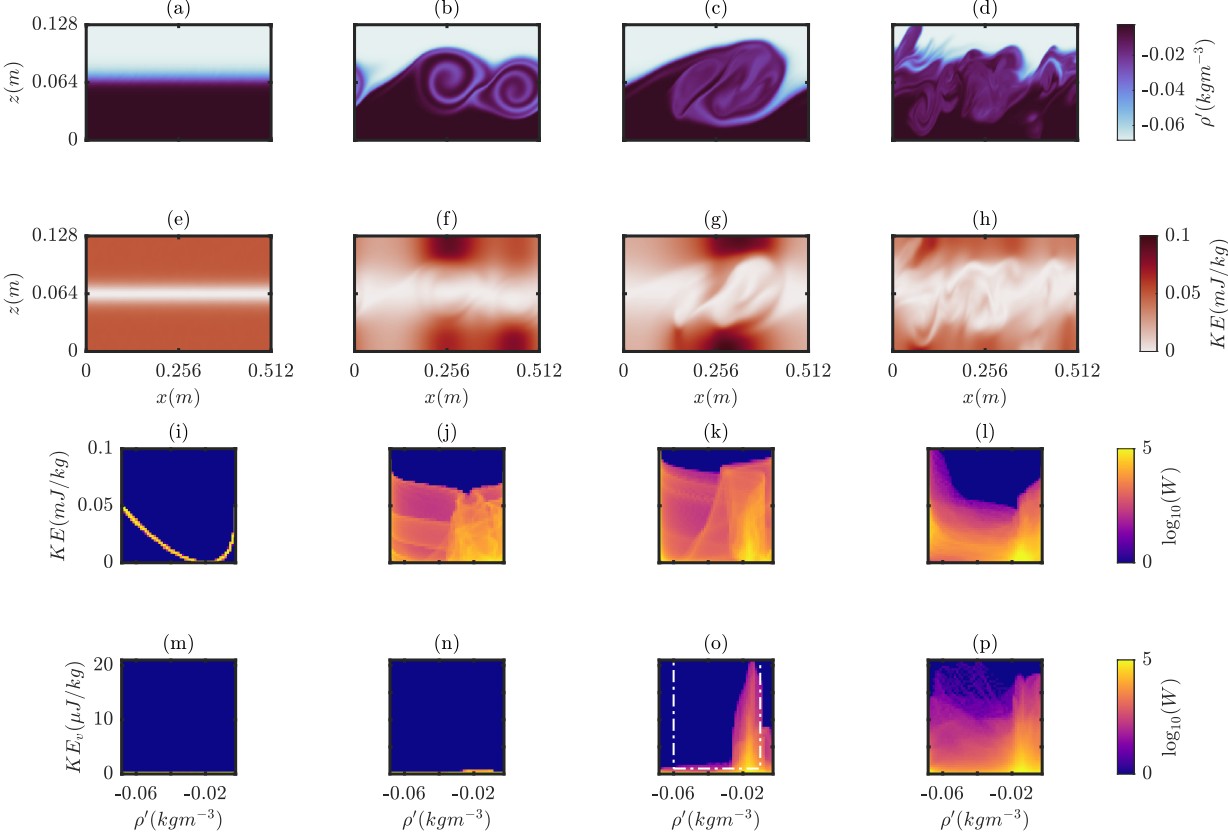

**Figure 5.** Time sequence of the cold shear instability, shown as density (a-d) and Kinetic Energy density (e-h) fields, and the USP for density vs KE density (i-l) and for density vs $KE_v$ (the KE density for just the $v$ component) (m-p). $t =$[5, 170, 225, 275]s.

As the flow begins to settle back towards the new steady state, kinetic energy at the intermediate densities falls, with the presence of the layer of intermediate density evident at $\rho' \approx -0.02$kgm$^{-3}$.

The USP region in which active mixing related to 3-D processes is identified in figure 5 o. This is where $KE_v$ is higher than any of the flow in early time steps, and with intermediate density. This region of the flow for $t = 225$s is isolated in figure 6. The transverse average $KE_v$ (figure 6 a) shows that this active 3-D mixing is occurring right at the core of the K-H billow, whilst there is considerable variability throughout the $y$ direction (figure 6 c, d). Peaks in $KE_v$ are particularly localised (figure 6 b), indicating the small scale processes at play responsible for the mixing. Such results are indicative of the importance of 3-D simulations for understanding these mixing processes. By isolating the fluid undergoing these processes, we can identify regions of the flow worthy of future study.



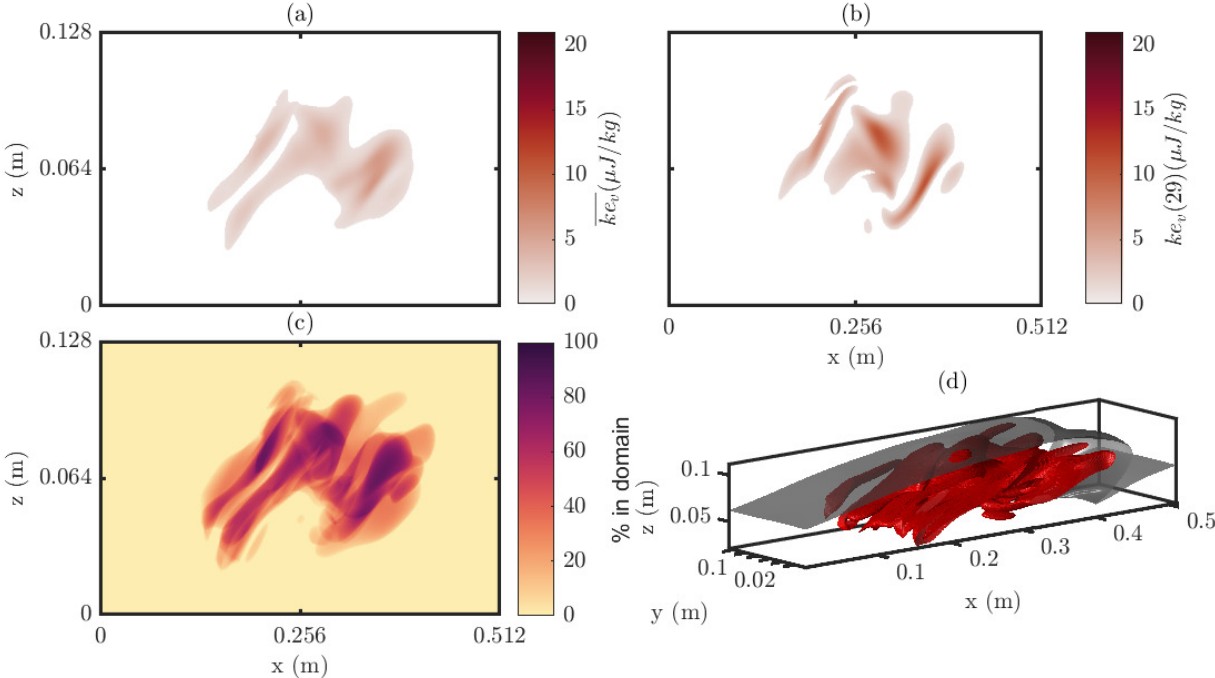

**Figure 6.** Regions of interest of the cold shear instability case at $t = 225$s, based on the region indicated in the white box of figure 5 o. Shown as the ROI in the transverse mean $\mathrm{KE}_v$ field (a), the $\mathrm{KE}_v$ $x, z$ slice most representative of the mean (b), the proportion of the $y$ domain within the ROI for the $(x, z)$ plane (c), and the boundaries of the ROI (red) with the $\rho' = -0.0454$ kgm$^{-3}$ isopycnal for reference (black) (d).

### 3.3 Cold shear instability with passive tracers

Two additional simulations were carried out with a passive tracer, $\Xi_i$, with different prescribed diffusivities. A less diffusive
case (case K-H Dye 1) had $\kappa_{\Xi_1} = 10^{-8}$m$^2$s$^{-1}$ (case K-H Dye 1) and a more diffusive case (case K-H Dye 2) had $\kappa_{\Xi_2} = 10^{-6}$m$^2$s$^{-1}$. These represent diffusivity approximately one order of magnitude lower and greater than the active temperature field, $\kappa_T = 1.4 \times 10^{-7}$m$^2$s$^{-1}$. A snapshot taken of the time step at which billows coalesce ($t = 225$s), reveals the emergence of fine structures in the K-H Dye 1 case within the billow core, and with sharp interfaces (figure 7 a), and a much more diffuse core, with blurred interfaces in the K-H Dye 2 case (figure 7 c). Such features manifest in the USP of the tracer against density
(figure 7 mid-row). A control with the active temperature tracer is presented (figure 7e), indicating the nonlinear relationship between density and temperature. The skew of this figure to the K-H Dye 1 case (figure 8d) shows which gradient is diffused faster, here indicating the density gradient is diffusing faster than the tracer, that is that the extremes of salinity, associated with the upper and lower layer fluids initially, now map onto a range of densities. Meanwhile, the skew of this figure to the K-H Dye 2 case (figure 7 f) shows the tracer gradient is diffusing faster than the density, that is that the extremes of density, associated



**Figure 7.** Late time step ($t = 225$s) plots for K-H Dye 1 case (left), the base case (centre) and K-H Dye 2 case (right). Top shows the tracer field (tracer for K-H Dye Case 1 and 2, temperature for base case), centre shows the density vs tracer USP, and lower row shows tracer vs $KE_v$.



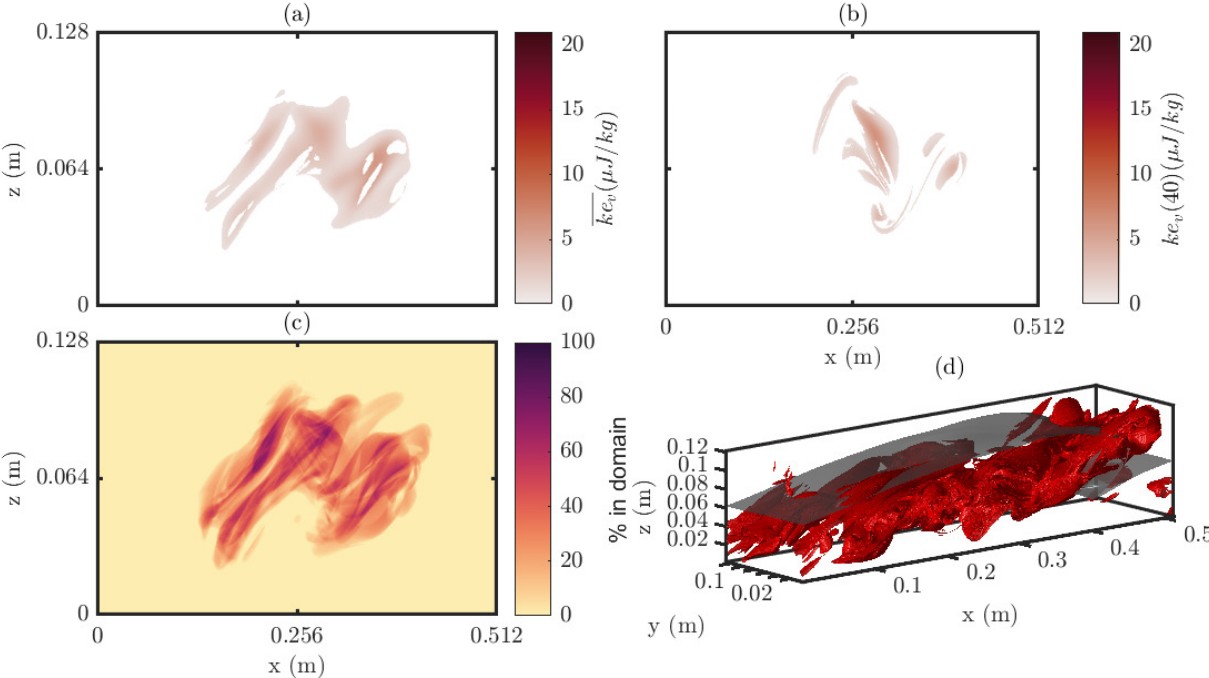

**Figure 8.** ROI plots as in figure 6 but for K-H Dye 1 case

with the upper and lower layer fluids, now map onto a range of densities, and indicating that if the fluids were sorted again by density, each layer would contain varying levels of the tracer (that is to say, the tracer has mixed).

Applying the same region of active 3-D mixing (based on the density of that fluid at the initiation of the simulation), as in figure 6 (see region in figure 7 h), produces figures 8 and 9. The mixing of the less diffusive passive dye indicates a pattern of highly localised, filamentous structures that are regions of active dye mixing (figure 8 b-d). In contrast, the diffusive passive dye is mixing in broader-scale structures (figure 9 c). However, the large-scale regions in which this passive dye is mixing is similar across both simulations (figure 8 a, c 9 a, c), indicating that there is an important interplay between the large scale flow structures (the K-H billow, and overturn of the isopycnals shown as the black isosurface) and filamentous fine-scale structures, across which the diffusive dye is able to diffuse. This result highlights the small scales at which diffusion plays an important role, only once the fluid and associated gradients are stretched to fine scales does the difference in diffusivity between dyes (and indeed the active tracer in figure 6) play any role.

## 4   Discussion

By pairing a kinematic measure (KE, $KE_v$, or $\Omega$) with a conserved tracer (density, temperature or passive dye), the role of both turbulence in stirring the fluid, and diffusive processes in irreversibly blurring density gradients are shown for three example



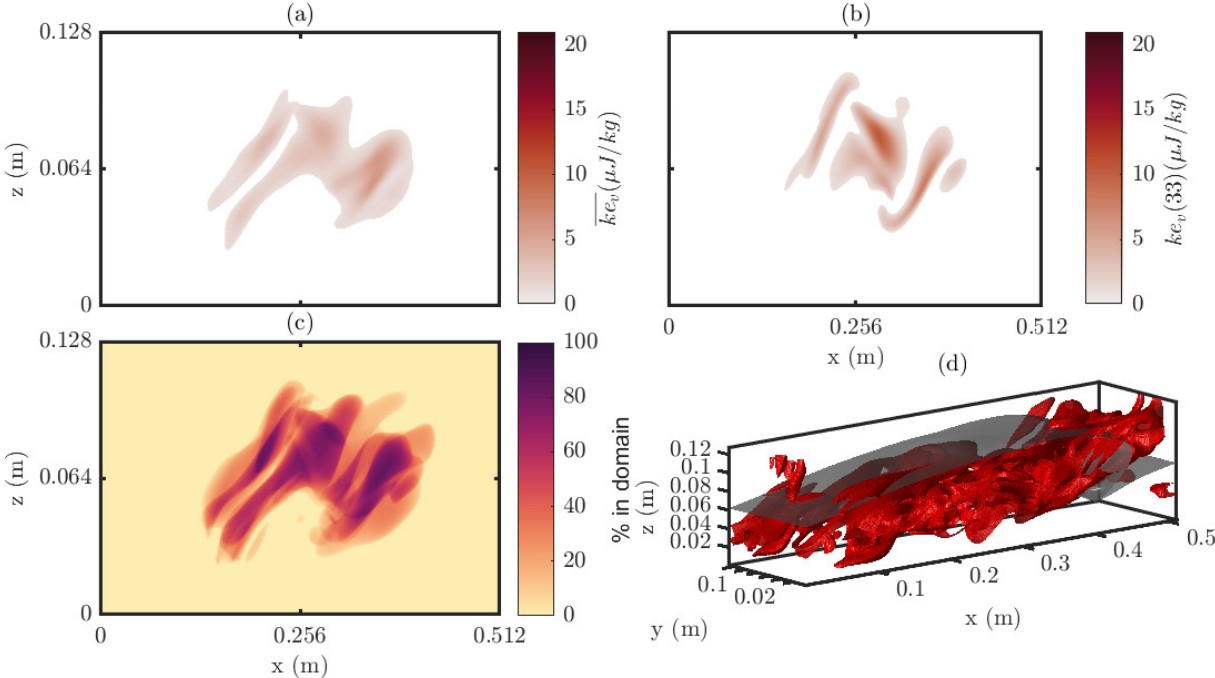

**Figure 9.** ROI plots as in figure 6 but for K-H Dye 2 case, for the ROI indicated by the white box in figure 7 h

stratified flows. This process has also been used to identify which fluid constituents are being mixed more effectively by a given

process, whether the low density upper layer fluid is being mixed into the pycnocline, or higher density, lower layer fluid mixed upwards, and the eventual fate of these fluids. In the case of the shoaling ISWs, informing whether mixed fluids are retained locally to the site of mixing, advected upslope, or away from the slope as an intermediate layer. Each of these questions is difficult to answer using existing bulk energetics models. The USP method has an additional advantage over existing methods for understanding in its ease of implementation, and its reliability, in cases where the model is under-resolved, or has unusual

bathymetry.

The results from shoaling ISWs reveal considerably different mixing regimes between breaking types, for example the horizontal extents of mixing between collapsing and plunging, where past studies have identified similar results between breaking types based on the energetics-based mixing measures alone. However, whilst it may be the bulk measure that is most useful in parameterising global-scale models, the locality of mixing, and transport of this fluid, which may carry with

it nutrients, heat (or thermal refuge), and sediment (depending on the local setting), is crucial to understand the effects of such processes on ecosystems. The features outlined by ROI in some shoaling ISW examples (e.g. figure 3 h) have striking similarity to the intermediate nepheloid layers identified by Bourgault et al. (2014), formed due to the shoaling of ISWs, and the interaction of sediment resuspension and mixing. Similar features have been observed in various locations, but due to their episodic nature, their formation mechanisms can be unclear (Schulz et al., 2021). Understanding these formation and evolution



processes help understand these features, and, may contribute to larger-scale sediment fluxes, and as a result carbon exports (Schulz et al., 2021). To investigate these processes, the USP method could be applied with an active sediment tracer.

For the cold shear instability, these results also reveal new insights into the dynamics of shear flows at low temperatures (below the density maximum). The shear flow presented could represent a cold river flow into a lake, and as such the position of the velocity gradient and temperature gradient are linked, whilst the density gradient position is offset. The result of this in

the shear flow is to produce asymmetric K-H features (e.g. figure 5b), and mixing bias towards one density (figure 5o). These asymmetric stratified shear flows have attracted recent attention following field observations (e.g Tu et al., 2020; Olsthoorn et al., 2023). Previous work on K-H billows has highlighted the role of 3-D instabilities beyond the point at which billows coalesce. Here, by isolating the fluid where these 3-D ($v$) flows are playing an active part in mixing, it shows that not only are 3-D processes important at this stage, but they remain within the coalesced billow region, and small scale (figure 6). Beyond

that stage, this $v$ component of the KE remains high around the active mixing region, but also increasingly across the entire flow. Furthermore, investigating the mixing of passive tracers is also not possible using energetics models, and has been presented here, identifying which densities the mixing of passive tracers is associated with, and the role of 3-D instabilities in the passive tracer mixing processes.

## 5   Conclusions

This new method identifies how the different regimes of mixing associated with different breaking types (and therefore with differing wave and slope characteristics) manifest themselves. ISWs breaking with surging behaviour produces a small mixing region, which is advected upslope, indicating transport of the mixed fluids, potentially to regions of high biological activity. In contrast, collapsing ISWs produce a large patch of actively mixing fluid over a long horizontal area, particularly with lower layer fluid mixed up into the pycnocline. Plunging ISWs, which initially appear to have similarities to collapsing ISWs, are

instead dominated by upper layers mixed down into the pycnocline, and are more horizontally constrained. Finally, ISWs undergoing fissioning produce active mixing over a long horizontal region, advected upslope in the form of quasi-turbulent dense pulses.

Temperature-stratified shear flow simulations in the cold, nonlinear EOS regime reveal important differences with a density-stratified shear flow. USP at early time steps immediately reveals the asymmetry of the distribution of density and kinetic

energy, an asymmetry which goes on to play an important role in the mixing induced by the shear instability. Once density gradients have been stretched, and transverse flow at medium-high densities forms, mixing occurs in an asymmetric manner, producing a late stage state with the layer of newly mixed fluid skewed towards medium-high densities. Simulations of the shear flow with passive tracers of varying diffusivity highlight the nature of flow structures associated with mixing fluid. At low diffusivities, filamentous structures in the tracer are observed, heavily associated with 3-D elements of the flow, whilst at

high diffusivity, tracer gradients are blurred. Both scenarios showing interplay between the dynamic drivers in large-scale flow features and diffusive effects at filamentous fine-scales once gradients are stretched. Such fine-scale structures are revealed by the new ROI plots based of USP. Overall, this new method has provided valuable insights into the evolution and dynamics

of sheared flows, and the role of temperature-based stratification within the nonlinear EOS regime, and diffusivity on mixing processes.

This method provides a method of interpreting and investigating mixing that is not blind, i.e. it invites the researcher to understand the ongoing processes in a way that is ignored by the relative black box of typical mixing methods. The tagging of particles based on variables provides a methodology that lies somewhere between a strictly Eulerian and strictly Lagrangian approach to understanding mixing. In the case of the passive tracer the method has been used to trace the motion of specific parcels of fluid. This could be further refined by introducing tracer at different times in the simulation into localized regions

of interest. Future work could also set up higher dimensional histograms (although these become increasingly difficult to visualise), and/or more complex selection criteria based on multiple quantities; effectively a language to classify aspects of the fluid motion.

*Code and data availability.* The interactive Matlab tool presented in this study can be found at https://github.com/HartharnSam/SPINS_USP. Data available from https://doi.org/10.25405/data.ncl.c.6704289 (prior to publication, reviewers can view the data at https://figshare.com/s/

3fa8f16a524806ecf84a).

*Author contributions.* S.H.E and M.S. performed, analysed, and curated data from the numerical simulations, produced the analysis code, and were responsible for study conceptualisation and design and writing the original draft. S.H.E was responsible for visualisation, M.C. and M.S were responsible for resource acquisition and supervision, and all authors reviewed and edited the manuscript, and contributed to funding acquisition.

*Competing interests.* The authors have declared that none of the authors have any competing interests.

*Acknowledgements.* This work was supported by the Natural Environment Research Council (NERC) funded ONE Planet Doctoral Training Partnership (S.H.E., grant number [NE/S007512/1]), and a Turing Global Fellowship (S.H.E). This work made use of the Rocket High Performance Computing service at Newcastle University and the Math Faculty Computing Facility at the University of Waterloo. Partial funding for travel was provided by the Natural Science and Engineering Research Council of Canada RGPIN-311844-37157. Exploratory

computations and discussions of these with Karem Abdul-Samad and Chris Subich are gratefully acknowledged.



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

**Appendix A:  Supplementary Figures**





**Figure A1.** Supplementary Material 1: Time sequence of surging wave shoaling process shown as $\rho'$ (top), $\Omega$ (middle), and USP for the same variables (bottom). t = [62, 70, 75]s





**Figure A2.** Supplementary Material 2: Time sequence of plunging wave shoaling process shown as $\rho'$ (top), $\Omega$ (middle), and USP for the same variables (bottom). t = [55, 60, 65, 70]s





**Figure A3.** Supplementary Material 3: Time sequence of fissioning wave shoaling process shown as $\rho'$ (top), $\Omega$ (middle), and USP for the same variables (bottom). t = [100, 108, 120, 136]s