# Peer review of "A new approach to understanding fluid mixing in process-study models of stratified fluids"

_EGUsphere, 2023_

## Author Comment (AC1)

RC1: 'Comment on egusphere-2023-1920', Anonymous Referee #1, 01 Oct 2023

Referee's Report on " A new approach to understanding fluid mixing in process-study models of stratified fluids " by Hartharn-Evans et al.

The authors proposed a tool for identifying stirring and mixing processes in stratified media. The approaches by Penney et al. (2020) and Grace et al. (2021) were extended and generalized.  Bivariate weighting histograms of fluid variables were used to identify mixing fluid. The weighting was related to the probability a volume of fluid has given properties in a 2D plot.

Two test cases were considered: shoaling internal solitary wave on bottom slope and shear instability in cold temperature stratified water. Unlike Penney et al. (2020) where the density-tracer pair was considered, the authors considered the density-kinetic energy pair. The results of a qualitative analysis of the diagrams are presented.

The developed approach is new and has a perspective used in different geophysical problems. The paper can be worth to be published in Nonlinear Processes in Geophysics after minor revision.

The authors thank the referee for their time and useful comments which have helped improve the manuscript. Please find below detailed responses to each of the reviewers comments in blue. Changes in the revised paper have been tracked, and a copy made available with these changes visible.

Specific comments

Introduction. It would be useful to include in the review other diagrams of the state and evolution of turbulence in a stratified medium (e. g. Caldwell (J. Geophys. Res. 88 C12, 19) and Gibson (J. Mar. Syst. 21, 1999)).

Thank you for these suggestions, a note about the historic use of graphical methods to understand turbulence in a stratified system has been added at the start of paragraph 4.

Fig. 4 caption Explain, please, what is shown in the plates of the middle row.

These plates show the regions of interest in the flow, with the fluid that meets the paired criteria shown in dark red. Words to this effect have been added to the caption of Figure 4.

L 24 Samgorinsky read as Smagorinsky

Thank you, this is now corrected

---

## Author Comment (AC2)

In the refereed paper, the authors suggest a method of determining how and where the mixing process occurs in water through the paired histograms approach of user-selected variables. Results of numerical simulations from previously published papers by the authors on shoaling internal solitary waves are used. The method allows researchers to identify regions of fluid in physical space that are undergoing mixing. Two specific cases are presented to illustrate the method: (i) shoaling internal solitary waves and (ii) a shear flow instability in water influenced by the nonlinearity of the equation of state. The method is also used to identify how the density and passive tracers are mixed within the core of the Kelvin–Helmholtz instability. By means of the suggested method becomes possible to identify how the different regimes of mixing associated with different types of wave breaking manifest themselves.

The paper is interesting and topical. Apparently, its further development will shed a lite on the onset of turbulence in fluid. It is well-written and well-illustrated; it can be recommended for publication in the NPG. I only have minor remarks that should be attended to before the paper publication.

*The authors thank the referee for their time and useful comments which have helped improve the manuscript. Please find below detailed responses to each of the reviewer's comments in blue. Changes in the revised paper have been tracked, and a copy made available with these changes visible.*

- It is not clear to me why the configuration of a flow shown in Fig. 2 was chosen such that it provides a minimum of nonlinearity. As well known, the nonlinearity diminishes when a pycnocline position is in the half-depth.

  *The authors note that figure 2b shows the configuration of the flow for the K-H instability, rather than for the ISW configuration, words to clarify this have been added to the caption of figure 2. In the shoaling internal solitary wave configuration, the pycnocline is not at the half-depth to avoid the minimum of nonlinearity. In this cold K-H case, the density is unusually offset from the half-depth due to the nonlinearity in the equation of state.*

- What type of internal solitary waves are realized in that configuration? Apparently, the detail of their shape is not important for this study but still, it would be good to mention what is the most relevant model that describes such solitary waves. Judging by the configuration of stratification and soliton shape shown in Fig. 3a), this is rather a Gardner soliton (see, e.g., Apel et al., JASA, 2007). What is the authors' opinion on this issue?

  *The waves in this configuration are strongly nonlinear solitary waves, which are well represented by the Dubriel-Jacotin Long equation (Turkington et al 1991 https://doi.org/10.1002/sapm199185293, Stastna & Lamb 2002 https://doi.org/10.1063/1.1496510). In our previous article (Hartharn-Evans et al 2022 https://doi.org/10.1017/jfm.2021.1049) for which these simulations were produced, the waves were in quantitative agreement with exact solitary waves computed by solving the fully nonlinear Dubreil–Jacotin–Long theory. Specifically, they are highly non-linear internal solitary waves of depression, and in some of the cases presented they are near the conjugate flow limit. More generally, it is important to note that these simulations are of the full stratified Navier Stokes simulations, not KdV based simulations.*

*When the pycnocline is close to the mid-depth (which is not the configuration away from topography in this study), it is true that Apel identifies that near the mid-depth wave trains often occur. Apel's observation would be relevant for future studies with a much shallower slope, for example for studies focusing on passage through the turning point.*

- There are a few typos that should be rectified. For example, the word "much" is repeatedly written in the Abstract. There is a typo in the word "nonlinearity" in the Abstract. And some more typos there are in the text.

*Thank you, the authors have corrected these specific typos, and completed a further proof read of the paper to catch any remaining typos.*